# Ward-based Goal-Directed Fluid Therapy (GDFT) in Acute Pancreatitis (GAP) trial: study protocol for a feasibility randomised controlled trial

Farid Froghi [1,2] Fiammetta Soggiu,[2] Federico Ricciardi [3]
Kurinchi Gurusamy [1] Daniel S Martin [1,4] Jeshika Singh,[5] Sulman Siddique,[1]
Christine Eastgate,[4] Maria Ciaponi,[4] Margaret McNeil,[4] Helder Filipe,[4]
Otto Schwalowsky-Monks,[4] Gretchelle Asis,[4] Massimo Varcada,[6]
Brian R Davidson[2]

¹Division of Surgery and Interventional Science, University College London, London, UK
²HPB and Liver Transplantation Surgery, Royal Free Hospital, London, UK
³Statistical Sciences, University College London, London, UK
⁴Critical Care Unit, Royal Free Hospital, London, UK
⁵Health Economics, PHMR, London, UK
⁶General and Emergency Surgery, Royal Free Hospital, London, UK

**Correspondence to**
Farid Froghi;
farid.froghi@nhs.net

## ABSTRACT

**Introduction** Acute pancreatitis is an inflammatory disease of the pancreas with high risk of developing multiorgan failure and death. There are no effective pharmacological interventions used in current clinical practice. Maintaining fluid and electrolyte balance is the mainstay of supportive management. Goal-directed fluid therapy (GDFT) has been shown to decrease morbidity and mortality in surgical conditions with systemic inflammatory response. There is currently no randomised controlled trial (RCT) investigating the role of GDFT based on cardiac output parameters in patients with acute pancreatitis in the ward setting. A feasibility trial was designed to determine patient and clinician support for recruitment into an RCT of ward-based GDFT in acute pancreatitis, adherence to a GDFT protocol, safety, participant withdrawal, and to determine appropriate endpoints for a subsequent larger trial to evaluate efficacy.

**Methods and analysis** The GDFT in Acute Pancreatitis trial is a prospective two-centre feasibility RCT. Eligible adults admitted with new onset of acute pancreatitis will be enrolled and randomised into ward-based GDFT (n=25) or standard fluid therapy (n=25) within 6 hours from the diagnosis and continuing for the following 48 hours. Cardiac output parameters will be monitored with a non-invasive device (Cheetah NICOM; Cheetah Medical). The intervention group will consist of a protocolised GDFT approach consisting of stroke volume optimisation with crystalloid fluid boluses, while the control group will receive standard care fluid therapy as advised by the clinical team. The primary endpoint is feasibility. Secondary endpoints will include safety of the intervention, complications, mortality, admission to intensive care unit, cost and quality of life.

**Ethics and dissemination** Ethics approval was granted by the London Central Research Ethics Committee (17/LO/1235, project ID: 221872). The results of this trial will be presented to international conference with interest in general surgery and acute care and published in a peer-reviewed journal.

**Trial registration number** ISRCTN36077283.

## Strengths and limitations of this study

► Adequate fluid therapy is a major determinant of outcomes in acute pancreatitis, however, there is no consensus on the rate, type and volume of fluids to be administered. Goal-directed fluid therapy (GDFT) based on stroke volume (SV) optimisation has been used after major surgery with a reduction in complications, however, delivering this intervention on the ward has been challenging without invasive monitoring.

► Novel development of non-invasive cardiac output monitoring (NICOM) technologies allows GDFT to be delivered in the ward based on haemodynamic parameters. This trial is the first GDFT randomised trial in acute pancreatitis using NICOM in the ward setting for SV optimisation.

► The golden period for intervention in acute pancreatitis is in the early phase of disease to prevent progression to severe inflammatory processes and complications. A major challenge in this trial is recruitment and delivering the intervention in the early hours of unplanned admission which is shared across trials in emergency care.

## INTRODUCTION

Acute pancreatitis is a sudden onset inflammatory process of the pancreas, with variable involvement of local or remote organ systems. The annual incidence of acute pancreatitis in the UK is approximately 30 per 100 000 of the population.[1] This equates to approximately 18 000 people developing acute pancreatitis every year in the UK.[1] There has been an increase in the incidence of pancreatitis in the last two decades.[2] Gallstones and excessive alcohol are the two main causes for acute pancreatitis in the UK.[3] Increasing age, male gender and lower

socioeconomic class are associated with a higher incidence of acute pancreatitis.[1]

The clinical manifestation of acute pancreatitis is believed to be caused by activation of inflammatory pathways either directly by the pathologic insult or indirectly by activation of trypsin, a protease which can break down the pancreas.[4] The average 60-day mortality associated with acute pancreatitis is 6.4%.[1] Deaths occur as a result of a massive fluid extravasation from the inflamed pancreas combined with a severe systemic inflammatory response in the early stage and mainly from local complications in the late stage.[5] The systemic complications occurring in the early phase result in poor oxygen delivery to the tissues and include worsening of pre-existing illnesses such as heart or chronic lung disease. Local complications which occur in the later phase include pancreatic pseudocysts, infected collections and pancreatic necrosis. These early and late complications are the major causes of the mortality associated with acute pancreatitis.[6 7] They are also largely responsible for the decreased health-related quality of life (HRQoL) and loss of work days following acute pancreatitis.[8] Thus, acute pancreatitis has a major effect on UK society with its complications having a significant impact on patients, their next of kin and their employers.

Various pharmacological interventions have been evaluated in acute pancreatitis, but none are in current clinical practice.[9] Supportive management in terms of maintenance of fluid and electrolyte balance remains the mainstay in the treatment of acute pancreatitis. Despite the key importance of fluid therapy, a recent systematic review has highlighted that there is a lack of information on the optimal fluid therapy in acute pancreatitis.[9 10] There has also been a decline in research in treatments for acute pancreatitis in general.[11]

The clinical presentation of acute pancreatitis varies from mild to severe life-threatening conditions. There is some evidence for aggressive fluid therapy with Hartmann's solution in those with severe disease. However, the rate and volume of fluid therapy is still debated for those with mild or moderate disease. Hence, high-quality randomised controlled trials (RCT) in fluid therapy for acute pancreatitis are necessary.

Goal-directed fluid therapy (GDFT) is a complex intervention in which intravenous fluid is given to optimise haemodynamic variables. This is achieved by stroke volume (SV) optimisation using a cardiac output (CO) monitor. RCTs have shown that GDFT decreases complications and mortality in other situations associated with a systemic inflammatory response.[12 13] Most of these trials however involved cardiac output monitoring for GDFT during surgery or in the intensive therapy unit (ITU) and not in a ward-based setting. For GDFT to be beneficial in acute pancreatitis, which involves a cascade of inflammatory events, therapy is likely to be most beneficial if commenced at the earliest opportunity to intervene following the onset of pancreatitis which would equate with the time of admission to hospital. There

has been one RCT on ward-based GDFT in patients with acute pancreatitis comparing the effect of GDFT with lactated ringers to normal saline on inflammatory response.[14] However, the trial was based on optimisation of blood urea nitrogen and not cardiac output parameters and failed to show a reduction in the inflammatory response or improved clinical outcomes. While other endpoints such as heart rate, urine output, haematocrit levels and central venous pressures have been suggested for fluid therapy in acute pancreatitis, it is the optimisation of intravascular volume with fluid therapy guided by cardiac output measures that has been previously shown to be effective in decreasing complications and mortality in major surgery.[15 16] There is currently no RCT investigating the role of ward-based GDFT using cardiac output parameters as target in patients with acute pancreatitis.

With the development of non-invasive cardiac output monitors such as Cheetah NICOM (Cheetah Medical, Maidenhead, Berkshire, UK), it is possible to measure the cardiac output in a ward setting.[17] Ward-based GDFT has the potential to decrease the early inflammatory response and hence complications related to acute pancreatitis which might result in decreased mortality and improved HRQoL. Reduced inflammation would also lead to a reduced intensive care unit (ICU) and hospital stay providing significant cost savings to the National Health Service (NHS), employers and care providers. This two-centre RCT will aim to assess the feasibility of guiding the initial 48 hours of intravenous fluid administration in patients with acute pancreatitis using ward-based GDFT.

## METHODS
### Study design and setting
The GDFT in Acute Pancreatitis (GAP) trial has been designed in accordance with the Standard Protocol Items: Recommendations for Interventional Trials guidelines as a two-centre feasibility RCT.[18] The study will primarily investigate the ability to recruit patients at the selected sites to a feasibility RCT of ward-based GDFT versus standard fluid therapy in patients with acute pancreatitis, the rate of withdrawal from GDFT protocol and the reasons for withdrawal from GDFT protocol. We will also assess the safety and practicality of ward-based GDFT and collect outcome measures which can be evaluated as endpoints for a subsequent multicentre study on efficacy. Indicative costs will be collected to inform a subsequent cost-effectiveness study. The trial outline is illustrated in figure 1 and the schedule of enrolment, interventions and assessments is shown in table 1.

### Participants
Adults (>16 years) admitted as an emergency with acute pancreatitis will be included. Acute pancreatitis must be confirmed by the international consensus criteria (box 1).[19] The exclusion criteria will be patients transferred for the management of complications of acute pancreatitis, requiring immediate admission to the ICU,

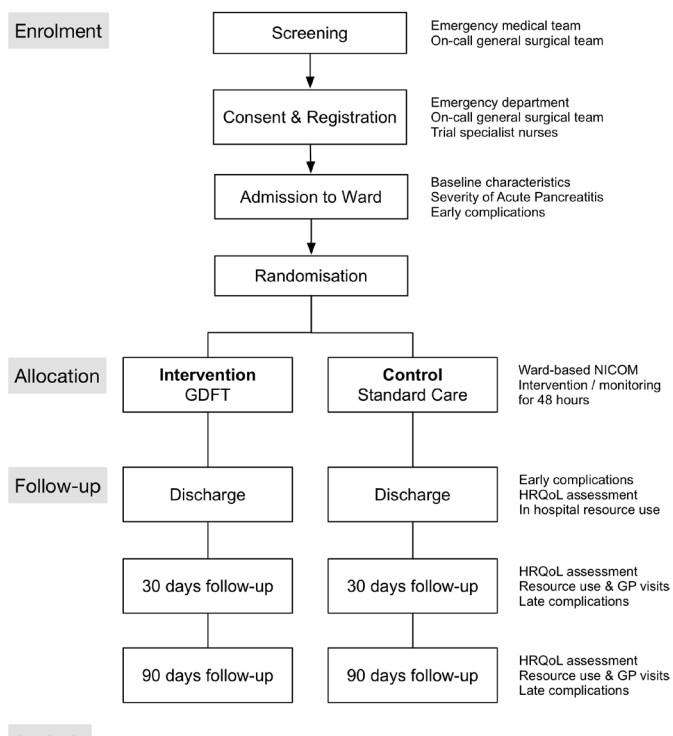

**Enrolment**

Screening — Emergency medical team / On-call general surgical team

Consent & Registration — Emergency department / On-call general surgical team / Trial specialist nurses

Admission to Ward — Baseline characteristics / Severity of Acute Pancreatitis / Early complications

Randomisation

**Allocation**

Intervention GDFT | Control Standard Care — Ward-based NICOM Intervention / monitoring for 48 hours

**Follow-up**

Discharge | Discharge — Early complications / HRQoL assessment / In hospital resource use

30 days follow-up | 30 days follow-up — HRQoL assessment / Resource use & GP visits / Late complications

90 days follow-up | 90 days follow-up — HRQoL assessment / Resource use & GP visits / Late complications

**Analysis** — **Study termination**

**Figure 1** Trial flow diagram. Patients are screened by the medical and surgical team in the emergency department for eligibility and referred for consent and registration. The trial nurse or surgical team provide trial patient information sheets (abbreviated and full versions) and gain informed consent. Trial nurses then randomise the patient to either GDFT or standard care and commence NICOM monitoring for 48 hours. Follow-up will be at the point of discharge, 30 and 90 days. GDFT, goal-directed fluid therapy; GP, general practitioner; HRQoL, health-related quality of life; NICOM, non-invasive cardiac output monitoring.

chronic pancreatitis in whom an acute exacerbation cannot be confirmed, a history of cardiac failure in the past 3 months and those unable to provide fully informed consent.

## Recruitment

Patients referred from the emergency department for admission through the emergency surgical team will be screened by the general surgical registrar on call. Due to the time-dependant nature of this trial, similar to most trauma and emergency surgery studies, a summarised and abbreviated patient information sheet will be provided for patients initially on arrival and they will be subsequently provided with an in-depth version. Informed consent will be obtained by a member of the clinical or research team trained in Good Clinical Practice.[20] The trial nursing staff responsible for instituting the trial intervention will be contacted within 4 hours of diagnosis. Intervention or standard of care will commence within 6 hours of the diagnosis for the next 48 hours of inpatient stay. Since the primary goal of this study is to assess the feasibility of delivering *ward-based* GDFT, patients requiring escalation of care and admission to ICU during the intervention

will only have continued cardiac output monitoring until completion of the 48 hours' period.

## Randomisation

This will be performed using 'Sealed Envelope', an internet-based randomisation system (www.sealedenvolpe.com). Eligible patients who have consented to take part in the trial will be 1:1 randomised to ward-based GDFT or standard care, stratified by site on admission, prior to ward transfer. The trial research nurse responsible (unblinded) will log into the database and create randomisation codes for both groups online. Information on randomisation will be stored in the case report form (CRF).

## Intervention

GDFT will be carried out for 48 hours. It can take up to 48 hours for the severity of pancreatitis to manifest and is considered the 'golden' period for interventions that may decrease severity.[21] GDFT will be based on a standard algorithm which uses the SV derived from non-invasive cardiac output monitoring (NICOM) using the Cheetah NICOM (figure 2). The fluid administration regimen will be as follows: maintenance fluid should be administered at 1.5 mL/kg/hour (based on ideal body weight) using a balanced crystalloid solution to be ensured that a regulated volume of fluid was given per hour in order to avoid fluid overload in this group. Previous studies have been criticised for excessive maintenance fluid prescribed in the GDFT group.[22] It was also felt that placing restrictions on the maintenance fluid prescription may influence decision-making in this group and affect the enthusiasm for clinicians to agree in patients participating in the study.

On admission SV is recorded and an initial bolus of 250 mL of intravenous fluid (balanced electrolyte solution) is given over 5–10 min. If there is a sustained rise in SV of greater than 10% for 15 min or more, this indicates fluid responsiveness and a repeat 250 mL bolus will be given. If there is not a rise in SV of greater than 10% then the patient is deemed fluid unresponsive and no further fluid boluses are administered. SV monitoring continues four hourly and if it decreases by more than 10% a further fluid bolus is administered as above.

## Control (standard care group)

The choice of fluid type, volume and rate of administration on the first 48 hours of admission for patients randomised to the control arm will be decided by the clinical team caring for the patient in order to mimic a real-world situation. Cardiac output monitoring variables will be measured every 4 hours on the standard care arm during the first 48 hours, however, the results will be blinded to the clinical team. The choice of fluid type, volume and rate will be recorded in detail. All other clinical, biochemical and HRQoL outcome measures will also be recorded for standard of care patients as outlined in the 'outcome measures' section.

**Table 1** Schedule of enrolment, interventions and assessments

| Procedures | Enrolment | Allocation | Postallocation | | | | | | | | | |
|---|---|---|---|---|---|---|---|---|---|---|---|---|
| **Time point** | **−t0** | **0** | **0 hour** | **12 hours** | **24 hours** | **48 hours** | **Discharge** | **7 days** | **30 days** | **90 days** | **Closeout** |
| **Enrolment** | | | | | | | | | | | |
| Eligibility screen | x | | | | | | | | | | |
| PIS given | x | | | | | | | | | | |
| Informed consent | x | | | | | | | | | | |
| Registration | x | | | | | | | | | | |
| Allocation | | x | | | | | | | | | |
| **Intervention** | | | | | | | | | | | |
| GDFT (intervention) | | | ◆———————————◆ | | | | | | | | |
| Standard care (control) | | | ◆———————————◆ | | | | | | | | |
| **Assessments** | | | | | | | | | | | |
| NICOM monitoring | | | ◆———————————◆ | | | | | | | | |
| Blood tests | | | x | x | x | x | | | | | |
| HRQoL | | | x | x | x | x | x | x | x | x | x |
| Complications | | | x | x | x | x | x | | x | x | x |
| Other outcomes | | | x | x | x | x | x | | x | x | x |

GDFT, goal-directed fluid therapy; HRQoL, health-related quality of life; NICOM, non-invasive cardiac output monitoring; PIS, patient information sheet.

**Box 1    International consensus criteria for acute pancreatitis**

**Acute pancreatitis diagnosis confirmed with two of the following three features:**

► Abdominal pain consistent with acute pancreatitis (acute onset of a persistent, severe, epigastric pain often radiating to the back).
► Serum amylase or lipase activity at least three times greater than the upper limit of normal.
► Characteristic findings of acute pancreatitis on contrast-enhanced CT (CECT) and less commonly MRI or transabdominal ultrasonography.

The Cheetah NICOM device (Cheetah Medical) is a CE (*Conformité Européene*) marked device which has been purchased for this study and training has been arranged for all staff on the equipment and the associated fluid

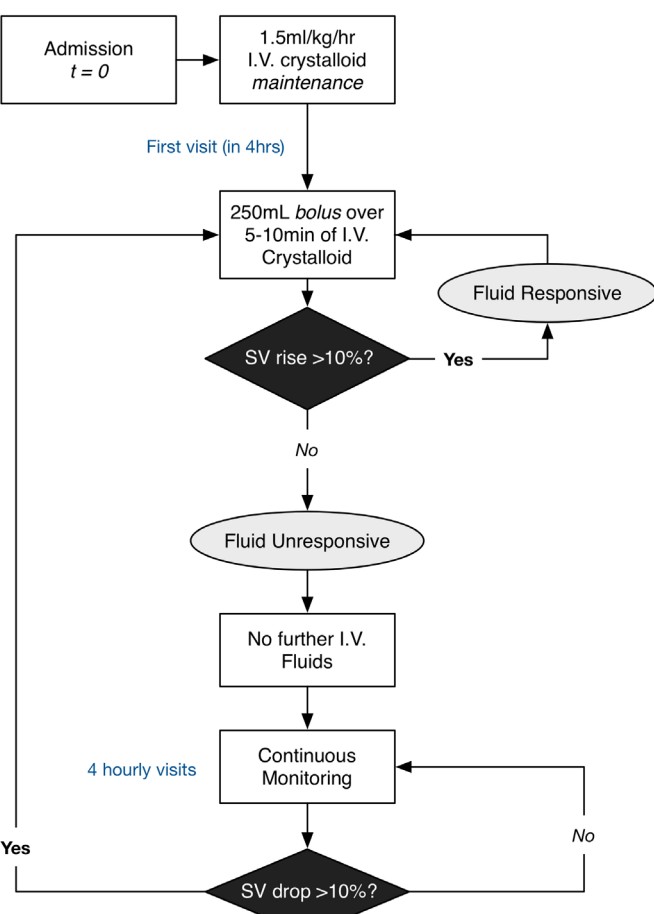

**Figure 2**    Goal-directed fluid therapy (intervention) protocol. Flow diagram of SV optimisation with maintenance fluid administration. Patients randomised to goal-directed fluid therapy (GDFT) will receive a 1.5 mL/kg/hour of intravenous crystalloid maintenance fluid and four hourly visits from the trial nurses. At the first 4-hour visit, a bolus of 250 mL crystalloid over 5–10 min will be administered. An SV rise of >10% would indicate fluid responsiveness and a further cycle of fluid bolus is administered until there is no SV rise >10%. At the next four hourly visits if SV has dropped by >10% the cycle is repeated. SV, stroke volume.

administration algorithm which will be a key component of the trial. The research team will follow hospital policy on signals for ICU outreach reviews and escalation of care to ICU. NICOM monitoring and outcome measures will be recorded in ICU for both control and GDFT arms.

### Blinding
It will not be possible to blind the research or clinical team during the first 48 hours of the study. However, the participants, outcome assessors of HRQoL and statisticians will be blinded to the groups. Patient blinding will be aided by NICOM monitoring of both intervention and control groups but performing GDFT in the intervention group alone. NICOM data from the control group will be collected at the same time points as the intervention group and will not be available to the treating clinicians.

### Outcome measures
The primary outcome of the trial will be an assessment of feasibility. In particular, the ability to identify and recruit patients at the selected sites to a study of acute pancreatitis. A recruitment rate of at least 30% over 17-month trial period will be deemed as successful. We will also assess the availability of study team for a condition presenting as an emergency, ability to randomise and commence ward GDFT within 6 hours of admission, completion rate of 48 hours of GDFT, rate of withdrawal (less than 20%) from GDFT protocol, reasons for withdrawal from GDFT protocol and proportion of complications in the two groups. Although the study is not powered to show differences in haemodynamic variables, we will analyse the haemodynamic data (such as SV and CO) at the end of the trial for patients in both groups to, at least, be able to conclude that we have optimised SV status of patients correctly according to the GDFT protocol drawn at the start of the trial. These data will be presented along with the main feasibility endpoints.

Several secondary clinical, surgical, biochemical, safety and quality of life outcomes will also be collected. Mortality will be assessed as well. Clinical outcomes will include severity of pancreatitis as assessed by Glasgow score, proportion of patients with severe acute pancreatitis, necrotising pancreatitis, infected pancreatic necrosis, requiring ITU stay, requiring renal replacement therapy, requiring ventilation, requiring surgical interventions for complications related to pancreatitis and incidence of positive blood cultures. Resource use data for health economic analysis on length of hospital stay, length of ICU stay, and number of days ventilated, time to return to pre-pancreatitis activities, number of workdays lost (in those who work) and costs (NHS and personal social services perspectives) will be collected. HRQoL will be assessed using the well-validated EuroQol-5 Dimension (EQ-5D) questionnaire on admission as baseline and subsequently on days 7, 30 and 90. All complications of pancreatitis including treatment-related adverse events (postradiological and surgical complications as classified by Clavien-Dindo Classification) and serious

adverse events related to intervention will be recorded up to discharge and at follow-up. Routine biochemical tests performed as part of clinical care including blood gases, liver function tests, clotting profile, renal function, amylase, full blood count and C-reactive protein (CRP) will be recorded on admission and daily up to 2 weeks and biweekly up to discharge. In addition, serum samples will be collected from participants in both groups on admission, 12, 24 and 48 hours' time points and stored at −80° for future analysis of biomarkers in acute pancreatitis.

## Follow-up

Participants will be followed for 90 days. All clinical and HRQoL outcomes will be measured up to discharge and specifically at 30 and 90 days by face-to-face or telephone follow-up. CRP will be measured at 24 and 48 hours and every 24 hours during the hospital stay as per routine clinical practice. Clinical outcomes on follow-up include cumulative hospital and ICU stay including readmissions, predefined complications including organ failure, severe pancreatitis, infected pancreatic necrosis, surgical interventions, development of sepsis, pulmonary oedema or acute respiratory distress syndrome as well as mortality. Resource use data include medications, planned and unplanned tests and procedures and use of allied healthcare professionals (eg, physiotherapy, occupational therapy). HRQoL will be assessed using a validated questionnaire (EQ-5D).

## Sample size calculation

Formal sample size calculations are not appropriate for a feasibility study, hence a convenient sample size of 50 patients, randomised equally to the two groups, has been chosen. Sample sizes between 24 and 50 have been recommended to estimate the SD required for a sample size calculation for a subsequent large RCT aimed at evaluating the cost-effectiveness of GDFT.[23]

## Patient and public involvement

Patients involved in the trial preparation were surprised that there had been no progress in effective treatment for acute pancreatitis and felt that new therapeutic approaches were required. Some had experienced late complications and felt that better early treatment may prevent these late complications. The trial protocol was presented and discussed with patients to ensure that patient-centred outcomes such as survival and quality of life were appropriately measured, and advice was sought on the conduct of the study. The burden of the intervention was assessed by patient representatives and was thought to be ethical and appropriate. The outcomes of the study will be disseminated to the participants after review and formulation by our study patient representative. We wish to thank our patient representatives for their contribution and participation in this process.

## Trial management structure

A Trial Management Group (TMG) chaired by the chief investigator and comprising leads at different sites, ICU research nurses, the surgical team representative, research fellows, statistician, qualitative researchers and health economist will meet fortnightly to discuss trial progress and address issues in recruitment and delivering the intervention. Trial progress will be reported to independent data monitoring committee (DMC) comprising experts in acute pancreatitis, medical statistics and clinical trials every 6 months.

All serious complications and any mortalities will be immediately referred to the sponsor and reported to the TMG and independent DMC. The trial will be stopped in case of two or more treatment-related suspected unexpected serious adverse reactions.

## Data collection and analysis

Recruitment data which include number of eligible patients presenting to the emergency department, number of patients screened, number of patients consenting and number of patients randomised are recorded in a recruitment log in the emergency department. A Consolidated Standards of Reporting Trials diagram will be presented to provide a detailed description of participant numbers at each time point during the trial.[24] A paper CRF is created for each participant enrolled into the study. The data collected on the paper CRF are then transferred to a secure online database using the Research Electronic Data Capture platform.

Since this is a feasibility study, all analyses other than recruitment rate and withdrawal rates should be considered exploratory. The two groups will be compared to ensure they have similar baseline characteristics (box 2) using means and SDs or medians and IQRs for continuous variables, as appropriate, and frequency counts and percentages for categorical variables.

For the primary outcome, that is, feasibility, the proportion of patients who consent to be randomised and the rate of withdrawal from GDFT protocol will be presented with a 95% CI. The median number of complications, graded by the Clavien-Dindo Classification, in each group will be presented. The proportions of people with

---

**Box 2    Baseline characteristics of patients**

The intervention group and the control group should have the following baseline characteristics:
► Age.
► Sex.
► Ethnicity.
► Body mass index (BMI).
► Time since onset of symptoms.
► Presumed cause of pancreatitis.
► Comorbidities.
► Baseline observations.
► Baseline renal function.
► Baseline blood gas parameters.
► Glasgow severity score on admission.
► Intravenous fluids prior to intervention.

complications between the two groups will be presented and compared using appropriate statistical test and 95% CI.

For the secondary outcomes, among patients who participate in the trial, all clinical and surgical outcome measures will be presented for each group separately using the mean, SD, median, minimum and maximum for continuous outcomes and using frequencies and proportions for categorical variables.

Quality of life will also be summarised for each group using mean profile plots over time. The mean difference in quality of life scores between the two groups at 7 days up will be presented with a 95% CI.

Kaplan-Meier survival estimates and plots will be presented to compare survival rates between the two groups.

All other secondary outcomes collected over time will be summarised for each group using mean profile plots. Mean differences for continuous outcomes and difference in proportion for binary outcomes shall be presented, with appropriate 95% CIs, at 30 and 90 days.

The number and nature of adverse events shall be reported for each group. No formal comparisons between the groups will be made and no hypothesis tests will be carried out. The results will inform us how sensitive the outcome measures are and, along with other information, will be used to determine the primary outcome of a subsequent large RCT. The results will also inform a sample size calculation for the primary outcome chosen.

### Assessment of feasibility

This is a feasibility study and one aspect of this feasibility evaluation is the ability to recruit patients with acute pancreatitis into a trial of this nature. According to hospital episode statistics, an in-house review of admissions for acute pancreatitis to the Royal Free Hospital over the last 3 years showed 107 cases per year. After excluding patients who were transferred from another hospital for tertiary care, those who required immediate ICU admission, those with chronic pancreatitis in whom an acute exacerbation could not be confirmed by internal consensus criteria, current or past cardiac failure, or unable to provide fully informed consent, 80 patients per year would be eligible for this trial from a single centre. With a second centre we would anticipate 120 suitable patients per year. We anticipate recruiting at least 30% of potential participants which equates to 36 participants annually and a recruitment time of 17 months to recruit the 50 patients.

For feasibility assessment, a recruitment log will be used to collect information on suitability and consent rate as well as availability of trials team for GDFT, consent, failure to commence ward GDFT, withdrawal with reasons and failure to complete the study. In case of lower than anticipated recruitment at 6 months, we will perform the following: (A) analyse the results of the qualitative research earlier and identify the reasons for poor recruitment (and take actions to resolve the problems) if we get the anticipated number of patients screened but the recruitment rate is lower than anticipated; (B) recruitment of additional centres if we do not get the anticipated number of potential participants screened but the recruitment rate is at least as good as the anticipated rate; (C) a combination of the above if the number of potential participants screened and the recruitment rate are lower than anticipated.

We will explore the reasons for participation and non-participation of eligible patients, and patients and clinicians' acceptability of the trial to assist in optimisation of recruitment strategies employed for the definitive trial. Non-participation can be related to how the clinical trial is presented to the patient, and how the patient assimilates this information. It is therefore important to understand how patients perceive information about potential participation and their experiences of receiving information relating to the trial. Interviews with a sample of eligible patients will explore patient perspectives of treatment, their understanding of the two treatments, reasons for taking part or refusing the trial and the acceptability of randomisation between the procedures. Interviews with clinical staff will explore their views about the trial, clinical equipoise and their understanding of the recruitment challenges. Semistructured interviews will be informed by a topic guide developed in conjunction with the TMG which includes patient representatives. Patient information sessions (recruiter meetings) will be audio recorded to examine how information is presented, and identify issues potentially affecting trial recruitment. Patient information sessions will be analysed at an early stage if there is poor recruitment and will inform the development of any additional training materials for recruitment for the definitive trial.

### Progress to full trial

The criteria to progress to a subsequent full trial will be determined quantitatively as (A) consent rate of at least 30%, the ability to recruit 50 patients to the study at the two sites over 17 months, (B) GDFT can be successfully performed within 6 hours of diagnosis of acute pancreatitis and can be continued until at least 48 hours after admission in a minimum of 80% of participants randomised to GDFT, and (C) the complication rate in the intervention group is not more than 10% higher than that of the control group at 90 days. A cut-off of 30% has been chosen for recruitment as this would be an achievable target in the definitive trial. A lower recruitment rate may indicate a lack of acceptability among clinicians and/or patients to participate in the definitive trial. In addition, a recruitment rate less than 30% is likely to make the subsequent definitive trial very expensive. A less than 30% recruitment may also highlight issues related to generalisability of the results, indicating a reduction in value of the subsequent definitive trial in terms of applicability in NHS.

### DISCUSSION

Acute pancreatitis is an inflammatory disease associated with a high mortality rate.[25] Adequate fluid resuscitation

has been identified as a key intervention in prevention of systemic complications, however, the debate over the type, rate or amount of intravenous fluids to be administered is ongoing.[26] With the advent of NICOM devices, delivering haemodynamic-guided fluid replacement is now possible in patients admitted to the general ward. The GAP trial is the first ward-based RCT of GDFT in acute pancreatitis. The trial will assess feasibility and safety of delivering this intervention in the ward and if successful, will progress to a multicentre trial to assess efficacy and cost-effectiveness.

## TRIAL STATUS

The GAP trial opened for recruitment on 8 January 2018 for a period of 18 months.

**Acknowledgements** We are grateful for the support of our emergency surgical colleagues Mr Olegunju Ogunbiyi, Mr Jonathan Knowles, Mr Rovan D'Souza, Mr Colin Hart, Mr Ahmad Hamad, the surgical registrars Ms Tanzeela Gala, Ms Karen Bosch, Ms Aliza Abeles, Mr Guillaume Lafaurie, Mr Tom Hanna, Mr Stephanos Pericleous, our emergency department lead Dr Jonathan Costello, without whom the conduct of this trial would not be possible. We thank our patient representative Ms Lucy Cotterell for their contributions. We gratefully acknowledge the financial support of the Wellington Hospital, London, in their HPB Surgery Fellowship funding for FF.

**Contributors** FF: manuscript preparation, mechanistic components of protocol design, trial and database management. FS: manuscript preparation, trial management. FR: statistical analysis plan. KG: major contribution to design and funding application. DSM: protocol design of GDFT regimes, trial management. SS: initial protocol developments and ethical approvals. CE, MC, OSM, GA, MM and HF: trial research nurses trained in delivering GDFT, data collection. JS: health economic analysis design. MV: participant enrolment planning through emergency surgery. BRD: principal investigator and grant holder, overall trial management, study design and review of manuscript. All authors read and approved the final manuscript.

**Funding** This paper presents independent research funded by the National Institute for Health Research (NIHR) under its Research for Patient Benefit (RfPB) programme (Grant Reference No PB-PG-0815-20002).

**Disclaimer** The views expressed are those of the author(s) and not necessarily those of the NHS, the NIHR or the Department of Health.

**Competing interests** None declared.

**Patient consent for publication** Not required.

**Ethics approval** The trial protocol (v2) has been reviewed and approved by the London Central Research Ethics Committee (REC Ref: 17/LO/1235, project ID: 221872). Informed consent will be obtained from eligible patients after screening by a member of the research or clinical team trained in Good Clinical Practice (GCP). The trial has been registered on ISRCTN on 9 April 2018 (http://www.isrctn.com/ISRCTN36677283). The results of this study will be presented to the national and international meetings with interest in the management of acute pancreatitis and prepared for publication in peer-reviewed journals with a readership in general surgery and critical care.

**Provenance and peer review** Not commissioned; externally peer reviewed.

**ORCID iDs**
Farid Froghi http://orcid.org/0000-0002-2895-117X
Federico Ricciardi http://orcid.org/0000-0002-4473-7691
Kurinchi Gurusamy http://orcid.org/0000-0002-0313-9134
Daniel S Martin http://orcid.org/0000-0001-6220-8235

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
