## [Reviewer comments · BMJ Open]

ARTICLE DETAILS

TITLE (PROVISIONAL)	Ward based Goal-Directed Fluid Therapy (GDFT) in Acute Pancreatitis (GAP trial): study protocol for a feasibility randomised controlled trial
AUTHORS	Froghi, Farid; Soggiu, Fiammetta; Ricciardi, Federico; Gurusamy, Kurinchi; Martin, Daniel S.; Singh, Jeshika; Siddique, Sulman; Eastgate, Christine; Ciaponi, Maria; McNeil, Margaret; Filipe, Helder; Schwalowsky-Monks, Otto; Asis, Gretchelle; Varcada, Massimo; Davidson, Brian

VERSION 1 – REVIEW

REVIEWER	Péter Hegyi Institute for Translational Medicine, University of Pécs. Pécs, Hungary
REVIEW RETURNED	12-Mar-2019

GENERAL COMMENTS	Acute pancreatitis has no specific treatment, therefore further studies are crucially needed. Indeed, adequate fluid therapy is a major determinant of outcomes in Acute Pancreatitis, however, there is no consensus on the rate, type and volume of fluids. The study is generally well designed and written. I have only few comments to add: 1) The authors mention in the introduction that there are not enough clinical trials concerning fluid therapy. I fully agree. However, the less amount of research on pancreatitis is unfortunately a general issue, which should be discussed PMID: 277761712) Recently an important summary of drug development and Trials in Acute Pancreatitis at the National Institute of Diabetes and Digestive and Kidney Diseases Workshop have been published (PMID: 30325856). The introduction would get benefits of discussing it as well.3) The results of the trial are crucially dependent on the time of intervention. The time period between the start of pain and intervention should be no more than 24 or 48h.4) The follow up plans of this trial should be discussed in more details.
--

REVIEWER	S. English Ottawa Hospital Research Institute
REVIEW RETURNED	16-Mar-2019

GENERAL COMMENTS	It is with pleasure that I have reviewed the submission by Forghi et al. entitled "Ward based Goal-Directed Fluid Therapy (GDFT) in Acute Pancreatitis (GAP trial): study protocol for a feasibility randomised controlled trial". In their manuscript, the authors detail
--

the methodology they have undertaken to complete a 50-patient, single blind (outcome assessor), feasibility RCT comparing 2 fluid resuscitation strategies - a goal-based strategy driven by optimizing SV as measured by a non-invasive cardiac output monitor (the intervention) compared to 'standard care' - during the first 48h of admission to a hospital ward for acute pancreatitis on outcomes including mortality, need for ICU admission, QOL (using EQ5D) and costs. I wish to commend the authors for their undertaking – early intervention studies that begin in the ER and continue on the ward are no easy undertaking – their timelines are ambitious as are their intervention goals given the number of clinical teams likely to be involved in the care of their participants. This is the perfect scenario in which a pilot RCT will help inform the conduct of the future larger trial powered to detect a clinically meaningful difference between the 2 groups. Kudos to the authors for initiating this process. A feasibility pilot RCT provides opportunity to, among other things,; 1) demonstrate ability to recruit the right patient population and in a time-frame conducive to being able to feasibly conduct the larger trial, 2) deliver the intervention as intended resulting in difference in management between the 2 study groups, and 3) be able to reliably assess the outcome you believe will be affected by the intervention. My greatest concern as it relates to this work is that although the authors make use of much of the potential benefits of such a pilot RCT, as written in this manuscript, it seems they may have overlooked #2. It is not clear to me how the authors intend to demonstrate the fidelity of their intervention or that there was in fact 'separation' between the 2 groups – that is, if there is no difference in the outcome measured how will the authors be confident that they are not erroneously concluding that the intervention made no difference rather than the fact that there was no difference in the management of the 2 groups? This may be particularly challenging in this study, where the intervention is not blinded – it is unclear how the authors intend to minimize the significant risks of bias inherent with such a design. It would be helpful for the reader to better understand the following:

1. The role of the 'research nurse' is unclear – it appears that this person (line 177 p.7) is 'unblinded' to the randomization schedule – this suggests a significant issue with allocation concealment and a significant threat to the trial result interpretability. If indeed someone who is not involved in the recruitment/intervention etc (eg trial statistician) was not charged with creating the randomization schedule, what safeguards were employed to counter this potential threat?
2. Randomization will be 1:1, stratified by center, - are blocks being used?
3. Although the authors stipulate that this pilot will “collect outcome measures which can be evaluated as end points for a subsequent multi-centre study on efficacy” – more clarity on the authors' thoughts as to potential primary outcomes of such a future trial would help in understanding the “feasibility metrics” of the pilot. For example, the necessary recruitment rate to conduct a future 200 patient trial is likely to be much different than a recruitment rate that would be necessary to demonstrate the feasibility of conducting a 2000 patient trial.
4. Related to clinical outcome – it would seem unlikely that mortality is the intended primary outcome of a future trial for a number of reasons including that 'severe pancreatitis' is largely excluded from this study and thus event rates would be extremely

	low. Much emphasis on mortality is made in the intro and again in the discussion. Given that this is presumably unlikely to be future trial primary outcome and to help the reader understand the 'biologic plausibility' of their intervention, a better understanding of the expected benefits (ie outcome to be measured in future trial) would be helpful. 5. In the intervention group, in addition to 'goal-directed therapy' this group appears to also be receiving a 'maintenance' IV fluid that isn't protocolled for the control group (line 188 p. 8)? Better justification for this difference in management which doesn't appear to be goal-directed, would be welcomed. 6. How do the authors intend to distinguish between volume overload that may be related to the intervention from acute lung injury which is common in this patient group? In general, a better sense of a priori defined safety outcomes and how they are defined/adjudicated would be helpful. This is especially important since the authors state that a difference of even 10% between the groups would be sufficient reason not to proceed. 7. According to the manuscript, this study began recruiting nearly 16 months ago – given the 17 month predicted timeline, reassurance that the pilot isn't already complete is necessary. Thank you again for the opportunity of reviewing this submission. I have enjoyed reading it and sincerely hope that my suggestions are helpful in making this an even stronger manuscript.
--	---

VERSION 1 – AUTHOR RESPONSE

Reviewer: 1

Reviewer Name: Péter Hegyi

Institution and Country: Institute for Translational Medicine, University of Pécs. Pécs, Hungary

Please state any competing interests or state 'None declared': None

Please leave your comments for the authors below

Acute pancreatitis has no specific treatment, therefore further studies are crucially needed. Indeed, adequate fluid therapy is a major determinant of outcomes in Acute Pancreatitis, however, there is no consensus on the rate, type and volume of fluids. The study is generally well designed and written. I have only few comments to add:

1) The authors mention in the introduction that there are not enough clinical trials concerning fluid therapy. I fully agree. However, the less amount of research on pancreatitis is unfortunately a general issue, which should be discussed PMID: 27776171

Added. Many thanks. Line 138.

2) Recently an important summary of drug development and Trials in Acute Pancreatitis at the National Institute of Diabetes and Digestive and Kidney Diseases Workshop have been published (PMID: 30325856). The introduction would get benefits of discussing it as well.

Many thanks. We have discussed UK guidelines for acute pancreatitis.

3) The results of the trial are crucially dependent on the time of intervention. The time period between the start of pain and intervention should be no more than 24 or 48h.

We agree that the timing is crucially important. As you will be aware, patients who develop acute pancreatitis have variable timing with respect to their onset of pain. We will record the timing of onset of the pain and consider this during the analysis of the results. However, the trial has been designed to reflect the possibilities of intervention once the patient presents and this would be the first 48 hours after admission to the hospital.

4) The follow up plans of this trial should be discussed in more details.

Done. Line 294.

Reviewer: 2

Reviewer Name: Shane English

Institution and Country: Ottawa Hospital Research Institute

Please state any competing interests or state 'None declared': none

Please leave your comments for the authors below

It is with pleasure that I have reviewed the submission by Forghi et al. entitled "Ward based Goal-Directed Fluid Therapy (GDFT) in Acute Pancreatitis (GAP trial): study protocol for a feasibility randomised controlled trial". In their manuscript, the authors detail the methodology they have undertaken to complete a 50-patient, single blind (outcome assessor), feasibility RCT comparing 2 fluid resuscitation strategies - a goal-based strategy driven by optimizing SV as measured by a non-invasive cardiac output monitor (the intervention) compared to 'standard care' - during the first 48h of admission to a hospital ward for acute pancreatitis on outcomes including mortality, need for ICU admission, QOL (using EQ5D) and costs.

I wish to commend the authors for their undertaking – early intervention studies that begin in the ER and continue on the ward are no easy undertaking – their timelines are ambitious as are their intervention goals given the number of clinical teams likely to be involved in the care of their participants. This is the perfect scenario in which a pilot RCT will help inform the conduct of the future larger trial powered to detect a clinically meaningful difference between the 2 groups. Kudos to the authors for initiating this process.

We wish to thank you for your kind comments and words of encouragement.

A feasibility pilot RCT provides opportunity to, among other things:

- 1) demonstrate ability to recruit the right patient population and in a time-frame conducive to being able to feasibly conduct the larger trial,
- 2) deliver the intervention as intended resulting in difference in management between the 2 study groups, and
- 3) be able to reliably assess the outcome you believe will be affected by the intervention.

My greatest concern as it relates to this work is that although the authors make use of much of the potential benefits of such a pilot RCT, as written in this manuscript, it seems they may have overlooked #2. It is not clear to me how the authors intend to demonstrate the fidelity of their intervention or that there was in fact 'separation' between the 2 groups – that is, if there is no difference in the outcome measured how will the authors be confident that they are not erroneously concluding that the intervention made no difference rather than the fact that there was no difference in the management of the 2 groups? This may be particularly challenging in this study, where the intervention is not blinded – it is unclear how the authors intend to minimize the significant risks of bias inherent with such a design.

Many thanks. As mentioned, the pilot study is not powered to show any differences in clinical outcome measures between the two groups. Hence any differences in clinical outcomes would not be reliable. The fluid management of patients with acute pancreatitis is highly variable between clinicians and centres. A goal-directed regime based on haemodynamic measures on the ward has never been subject to a randomised trial in acute pancreatitis patients.

An important and valid point regarding fidelity of treatment. We will ensure all fluid administration – including timing, type, volume and rate are recorded in the two groups. Patients in both groups (control and intervention) will have haemodynamic monitoring every four hours using the Cheetah NICOM monitor.

We will analyse the haemodynamic data at the end of the trial for patients in both groups to, at least, be able to conclude that we have optimised SV status of patients correctly according to the GDFT protocol drawn at the start of the trial. This data will be presented along with the main feasibility end-points.

It would be helpful for the reader to better understand the following:

1. The role of the 'research nurse' is unclear – it appears that this person (line 177 p.7) is 'unblinded' to the randomization schedule – this suggests a significant issue with allocation concealment and a significant threat to the trial result interpretability. If indeed someone who is not involved in the recruitment/intervention etc (eg trial statistician) was not charged with creating the randomization schedule, what safeguards were employed to counter this potential threat?

Allocation concealment is not a threat in this case. The randomisation is performed by computer generated software – sealed envelope – and there is allocation concealment, that is to say, the trial nurse is not aware to which group the next patient will be randomised to.

There is no blinding of trial nurses and clinical teams for the following reasons:

It is impossible to blind the trial nurses as they are instituting the monitoring and delivering the goal-directed fluid therapy protocol.

The clinical team cannot be blinded as they would be delivering the standard care – however, they are blinded to the readings of the NICOM monitor during the 48-hour period.

2. Randomization will be 1:1, stratified by center, - are blocks being used?

Block randomisation is not being used.

3. Although the authors stipulate that this pilot will “collect outcome measures which can be evaluated as end points for a subsequent multi-centre study on efficacy” – more clarity on the authors' thoughts as to potential primary outcomes of such a future trial would help in understanding the 'feasibility metrics' of the pilot. For example, the necessary recruitment rate to conduct a future 200 patient trial is likely to be much different than a recruitment rate that would be necessary to demonstrate the feasibility of conducting a 2000 patient trial.

In this feasibility study we will assess acceptability of the intervention by patients and professionals, practicality and system integration as well as safe and timely delivery of the intervention to then allow the decision to be made to proceed with a controlled efficacy trial at a larger scale. The feasibility metrics for the pilot are discussed at length with regards to recruitment rate, withdrawal rate, and complications relating the intervention.

The primary outcome for efficacy testing in a larger randomised clinical trial would then be a clinically relevant end-point such as progression to severe pancreatitis or persistent organ failure or mortality.

4. Related to clinical outcome – it would seem unlikely that mortality is the intended primary outcome of a future trial for a number of reasons including that 'severe pancreatitis' is largely excluded from this study and thus event rates would be extremely low. Much emphasis on mortality is made in the intro and again in the discussion. Given that this is presumably unlikely to be future trial primary outcome and to help the reader understand the 'biologic plausibility' of their intervention, a better understanding of the expected benefits (ie outcome to be measured in future trial) would be helpful.

Severe pancreatitis is not excluded in this trial. Since, one of the centres in this trial is a tertiary referral centre for hepato-pancreatico-biliary disease, patients transferred from other hospitals for treatment of complications related to pancreatitis are excluded from this study as they would be outside the time frame for starting the intervention (within 6 hours of diagnosis).

Likely primary outcome for future trials would include complications such as progression to severe acute pancreatitis, SIRS, ARDS and renal failure and mortality.

5. In the intervention group, in addition to 'goal-directed therapy' this group appears to also be receiving a 'maintenance' IV fluid that isn't protocolled for the control group (line 188 p. 8)? Better justification for this difference in management which doesn't appear to be goal-directed, would be welcomed.

We left all fluid intervention at the discretion of the clinicians for the control group, in order to mimic a real-world situation. We felt that placing restrictions on the maintenance fluid prescription may influence decision making in that group and affect the enthusiasm for clinicians to agree in patients participating in the study. For the GDFT group we wanted to be sure that a regulated volume of fluid was given per hour in order to avoid fluid overload in that group. Previous studies have been criticised for excessive maintenance fluid prescribed in the GDFT group (Challand et al 2011).

6. How do the authors intend to distinguish between volume overload that may be related to the intervention from acute lung injury which is common in this patient group?

In general, a better sense of a priori defined safety outcomes and how they are defined/adjudicated would be helpful. This is especially important since the authors state that at difference of even 10% between the groups would be sufficient reason not to proceed.

If there is an excess of pulmonary complications in one or other of the groups it will be possible that this is due to the fluid intervention. However, this study being a feasibility study, it is not sufficiently powered to draw conclusions about such findings. If we were to find an excess of pulmonary complications in the GDFT group, it is certainly plausible this may be due to fluid overload.

7. According to the manuscript, this study began recruiting nearly 16 months ago – given the 17 month predicted timeline, reassurance that the pilot isn't already complete is necessary.

The study is currently recruiting. See line 413.

Thank you again for the opportunity of reviewing this submission. I have enjoyed reading it and

sincerely hope that my suggestions are helpful in making this an even stronger manuscript.

VERSION 2 – REVIEW

REVIEWER	S English Ottawa Hospital Research Institute, Canada
REVIEW RETURNED	22-Jul-2019

GENERAL COMMENTS	Thank you for the opportunity to re-review the revised submission of the GAP trial - a pilot RCT to measure feasibility of a ward-based EGDT intervention (compared to control) in acute pancreatitis. I appreciate the authors consideration of my previous comments and responses to same. Although most have either been addressed or explained in their response, I remain concerned about the lack of description in the re-submitted manuscript on how the fidelity of the intervention will be measured (or reported) and compared to control. The authors have addressed this briefly in their response provided but as this is unlikely to be read by the average reader of the published protocol inclusion of these details in the main manuscript would be beneficial. Similarly, the explanation provided for the justification of the 'maintenance' IV fluids in only the intervention arm should be included in the manuscript, particularly since it by its nature is not 'goal directed'. I have no additional concerns.
---

VERSION 2 – AUTHOR RESPONSE

Reviewer: 2

Reviewer Name: S. English

Institution and Country: Ottawa Hospital Research Institute, Canada

Please state any competing interests or state 'None declared': No relevant competing interests.

Please leave your comments for the authors below

Thank you for the opportunity to re-review the revised submission of the GAP trial - a pilot RCT to measure feasibility of a ward-based EGDT intervention (compared to control) in acute pancreatitis. I appreciate the authors consideration of my previous comments and responses to same.

Many thanks for your detailed review of the protocol. It has certainly made it a more complete paper.

Although most have either been addressed or explained in their response, I remain concerned about the lack of description in the re-submitted manuscript on how the fidelity of the intervention will be measured (or reported) and compared to control. The authors have addressed this briefly in their response provided but as this is unlikely to be read by the average reader of the published protocol inclusion of these details in the main manuscript would be beneficial.

A paragraph has been added to the manuscript to this effect. Line 238.

Similarly, the explanation provided for the justification of the 'maintenance' IV fluids in only the intervention arm should be included in the manuscript, particularly since it by its nature is not 'goal directed'. I have no additional concerns.

A paragraph has been added to the manuscript to this effect. Line 192 and 209.